# Barriers to and Enablers of the Inclusion of Micronutrient Biomarkers in National Surveys and Surveillance Systems in Low- and Middle-Income Countries

**DOI:** 10.3390/nu14102009

**Published:** 2022-05-10

**Authors:** Mari S. Manger, Kenneth H. Brown, Saskia J. M. Osendarp, Reed A. Atkin, Christine M. McDonald

**Affiliations:** 1International Zinc Nutrition Consultative Group, University of California, San Francisco, CA 94143, USA; christine.mcdonald@ucsf.edu; 2Department of Nutrition, Institute for Global Nutrition, University of California Davis, Davis, CA 95616, USA; khbrown@ucdavis.edu; 3Micronutrient Forum, Washington, DC 20005, USA; saskia.osendarp@micronutrientforum.org (S.J.M.O.); reed.atkin@micronutrientforum.org (R.A.A.); 4Departments of Pediatrics, Epidemiology and Biostatistics, University of California, San Francisco, CA 94143, USA

**Keywords:** micronutrient, biomarker, survey, surveillance

## Abstract

Including biomarkers of micronutrient status in existing or planned national surveys or surveillance systems is a critical step in improving capacity to promote, design, monitor, and evaluate micronutrient policies and programs. We aimed to identify the barriers to and enablers of the inclusion of micronutrient biomarker assessment in national surveys and surveillance systems, to identify the main challenges faced during the survey process, and to review experiences using existing platforms for micronutrient surveys. We conducted a series of key informant interviews with in-country and external representatives from six countries where national-level data on micronutrient status were collected in the past 5 years: Cambodia, Pakistan, Malawi, Uganda, Ghana, and Uzbekistan. Micronutrients associated with specific public health programs were always prioritized for inclusion in the survey. If funding, time, and/or logistics allowed, other considered micronutrients were also included. The most important and frequently reported barrier to inclusion of a more comprehensive panel of micronutrient biomarkers was inadequate funding to cover the laboratory analysis cost for all micronutrients considered at the planning stage. Government support and commitment was stressed as the most important enabling factor by all key informants. Advocacy for funding for micronutrient status assessment is needed.

## 1. Introduction

Increasing the availability, accessibility, and utilization of micronutrient status data would dramatically improve the ability to promote related public health and nutrition policies, as well as design, monitor, and evaluate micronutrient programs. Ultimately, investing in better data would yield healthier populations and safer programs and potentially result in cost savings. To increase the availability of micronutrient status data, biomarkers of micronutrient status will need to be included in planned national surveys or surveillance systems. However, many low- and middle-income countries (LMICs) lack high-quality, routinely collected data on micronutrient status. The Lancet Series on Maternal and Child Undernutrition Progress [1] drew attention to the vast data gap for biomarkers of micronutrient status, particularly for women of reproductive age (WRA), and called for renewed efforts and funding to close this gap.

While some biomarkers, such as hemoglobin, are routinely measured in health and nutrition surveys, status indicators for specific micronutrients such as zinc, folate, vitamin B12, thiamin, and vitamin D are less commonly assessed or are assessed infrequently. The WHO Vitamin and Mineral Nutrition Information System (VMNIS) is the most comprehensive and publicly accessible source of nationally representative data on micronutrient biomarker status globally [2]. With the caveat that more surveys may have been conducted but results were not made available, between 1988 and 2018, just 38% of LMICs reported data on iron status to the WHO VMNIS for preschool children (PSC), 56% reported data on vitamin A status, 16% reported on serum zinc, 6% reported on vitamin D, and 5% reported on vitamin B12 [3]. Among nonpregnant WRA, only 17% of LMICs reported data on folate status [3]. Very few LMICs have published data on thiamine, riboflavin, or selenium status among PSC and WRA [4,5]. In addition to this paucity of data, the information that is available is often outdated; for iron status, the average year for countries only conducting one survey between 1998 and 2018 was 2005, and the average number of years between surveys for those countries where two surveys were conducted was 9 years [3].

The predominant approach for collecting national-level data on micronutrient status is to conduct standalone nutrition and/or micronutrient surveys. In some countries, biomarkers of micronutrient status may also be included as part of a national surveillance system [6]. Although significant effort has gone into publishing recommendations on technical issues related to micronutrient biomarker assessment [7,8,9,10,11,12], less is known about the main factors that either prevent or enable countries to conduct comprehensive, population-level micronutrient status assessments.

To this end, we carried out a series of key informant interviews to identify barriers and enabling factors behind the inclusion of biomarkers of micronutrient status in national surveys and surveillance systems. As secondary objectives, we also aimed to identify the main challenges faced during collection, processing, transport, storage, and analysis of biological specimens, as well as to review experiences and perspectives regarding the inclusion of micronutrient biomarkers as part of Demographic and Health Surveys (DHS) as an example of an existing platform that could be leveraged for the assessment of micronutrient biomarker data.

## 2. Materials and Methods

### 2.1. Country Selection

Due to budget and time constraints, it was decided a priori to include six countries in this study. The countries were selected on the basis of the following characteristics: (1) a national nutrition survey or surveillance was carried out in the past 5–10 years with emphasis on surveys in the past 5 years. These countries were identified from a global inventory of completed and upcoming surveys which was compiled and maintained by the International Zinc Consultative Group (IZiNCG) over the previous year; (2) countries reflected a variety of geographical regions; (3) a variety of technical support agencies were represented across countries; (4) at least one survey collected micronutrient biomarker data as part of a DHS. Technical support agencies were defined as governmental (e.g., US Centers for Disease Control (CDC)) or private organizations (e.g., GroundWork) providing external technical support to countries.

### 2.2. Key Informant Selection

Country key informants were identified with the help of the main agency that provided technical support to the design and implementation of the particular survey. The aim was to interview one in-country representative and one representative from the lead external support agency, primarily CDC and GroundWork, for each country. In addition to the country-specific internal and external informants, key informants from the US Centers for Disease Control (CDC), UNICEF, and ICF Macro (the lead implementing partner of the DHS Program) were invited to provide a global perspective.

### 2.3. Interviews

The primary interview guide for the in-country representatives was developed with input from representatives of the Micronutrient Forum and the IZiNCG Steering Committee. The guide followed the different stages of conducting a survey, i.e., proposal writing and fundraising, the planning phase, the implementation phase, laboratory analysis, data analysis, and results dissemination. For interviews with representatives from external agencies that provided funding or technical assistance to the surveys, the primary interview guide was modified slightly to encompass specific prompts about the role of the external agency and how it became involved in the survey. Furthermore, barriers and enablers were discussed from their experience working with several countries, and questions included steps the international community could take to expand the inclusion of micronutrient biomarkers in national surveys and surveillance systems.

The interviews were conducted between September and November 2019. The main questions of the interview guide, i.e., without interviewer probes, were shared electronically with the informants 1 week prior to the interview. All interviews were conducted by the first author (M.S.M.), with a second author (C.M.) participating in two of the interviews. The interviews were conducted by conference call and were approximately 90 min in duration. Due to time zone constraints, in-country key informants and supporting agency informants were typically interviewed separately rather than in pairs. All interviews were recorded after obtaining informed consent from the informants. The study was approved by the Institutional Review Board of the University of California, San Francisco.

### 2.4. Data Analysis

Detailed notes were taken from the recorded interviews. For each key informant interview, major themes and illustrative excerpts arising for each question were recorded. In the case of surveys for which there were two key informants, themes and excerpts were collated for both informants for that survey. Following this, a thematic analysis approach [13] was used to organize themes and excerpts into predetermined categories in line with research objectives. The categories were further reviewed and refined by going back to the detailed notes. Barriers reported by the respondents were classified according to importance, as emphasized and/or ranked by the respondent and the frequency (per country) with which the barrier was reported. The perspectives of external technical support agencies were compared and contrasted with country perspectives.

## 3. Results

### 3.1. Overview

The country surveys and surveillance systems included in this study were conducted in Cambodia [14], Malawi [15], Pakistan [16], Uzbekistan [17], Ghana [18], and Uganda (Table 1). A total of 12 interviews with 13 key informants were conducted. In four countries, key informants from both the domestic and the external lead agencies were interviewed. For two countries, only one key informant was interviewed because the key informant either represented the lead domestic agency which was also the lead technical assistance agency (Aga Khan University, Karachi, Pakistan) or was external but permanently based in the country (Institut de Recherche pour le Développement, Lyon, France). One key informant each from CDC, UNICEF, and ICF Macro was interviewed to provide a global perspective.

### 3.2. Characteristics of the Included Surveys

#### 3.2.1. General Characteristics

The surveys were initiated by the country’s Ministry of Health in Malawi, Uganda, and Uzbekistan. In Cambodia and Ghana, representatives from UNICEF advocated for the need for survey data. In Pakistan, a local champion played a critical role in convincing donors to support the survey. The characteristics of the six included surveys are shown in Table 1. UNICEF was the primary donor in three surveys. External technical support was provided by CDC and GroundWork in two surveys each (Malawi and Uganda, and Ghana and Uzbekistan, respectively). In Cambodia and Pakistan, the Institut de Recherche pour le Développement, France (IRD) and Aga Khan University provided technical support, respectively. Three countries conducted standalone nutrition surveys including micronutrient biomarkers, two countries assessed micronutrient biomarkers in association with a DHS survey, and one country collected micronutrient biomarkers as part of a National Panel Survey. The level of representativeness was urban–rural and/or macro-region in four surveys and district or micro-region in two surveys.

#### 3.2.2. Micronutrient Biomarkers and Laboratories

Hemoglobin, serum ferritin, C-reactive protein (CRP), and α-1-acid glycoprotein (AGP) were assessed in both PSC and WRA in all surveys (Table 2). Serum soluble transferrin receptor (sTfR) was assessed as an additional biomarker of iron status in four of the surveys. Serum folate was assessed in all six surveys, but only among WRA in three surveys. Red blood cell (RBC) folate was assessed in three surveys, in Malawi, Pakistan, and Uganda. Vitamin B12 was assessed in all six surveys, but only among WRA in three surveys. Serum retinol was assessed in five surveys, but as a subsample together with MRDR in three surveys. Retinol-binding protein (RBP) was assessed in four surveys; in three of these surveys, it was accompanied by a measurement of serum retinol in a subsample, and, in one of these surveys, it was the only biomarker of vitamin A status. Five surveys assessed urinary iodine among PSC, but also among WRA only in Malawi, Uganda, and Uzbekistan. Ghana did not include urinary iodine assessment because iodine status was recently assessed in a nationally representative survey.

Biomarkers of the following micronutrients were collected in three or fewer surveys: zinc, vitamin D, calcium, thiamine, and selenium. Plasma/serum vitamin D and serum calcium were assessed together in two surveys, while RBC thiamine diphosphate and plasma selenium were assessed in one survey each. None of the surveys measured biomarkers of riboflavin or niacin status; however, assessment of urinary niacin metabolites was discussed as a post hoc analysis for the Malawi survey.

HemoCue 301 was used for analysis of hemoglobin in all six surveys, using capillary blood in Cambodia and Ghana, and venous blood in the remaining surveys. Blood samples were analyzed domestically in two surveys and were fully or partially exported for analysis in four of the surveys, with countries shipping samples to 2–4 international laboratories per survey. The international laboratories used were VitMin Laboratory (Willstaett, Germany), Peking University (Beijing, China), Mahidol University (Bangkok, Thailand), Institute of Nutrition of Central America and Panama (INCAP; Guatemala City, Guatemala), CDC (Atlanta, GA, USA), United States Department of Agriculture Agricultural Research Service Western Human Nutrition Research Center (SDA ARS WHNRC, Davis, CA, USA), University of Wisconsin (Madison, WI, USA), University of California, San Francisco (Oakland, CA, USA), and National Institute of Nutrition (Hanoi, Vietnam).

### 3.3. Barriers to the Inclusion of Micronutrient Biomarkers

Barriers to the inclusion of micronutrient biomarkers focused on including a more comprehensive panel of micronutrient biomarkers. However, throughout the in-country interviews and particularly in the overview interviews, barriers to assessing micronutrient status at all were also discussed. All key informants reported the selection process as starting with the micronutrients that were associated with large-scale nutrition programs in the country that directly or indirectly aimed to improve micronutrient status (e.g., vitamin A supplementation programs, salt iodization programs). However, all survey committees considered a more comprehensive set of micronutrient biomarkers than what was ultimately included in the final survey.

Barriers to inclusion of micronutrient biomarkers are presented in Table 3 according to the frequency with which the barriers were reported, as well as the importance placed on the factor as reported by the respondents.

#### 3.3.1. Financial Barriers

Given that all surveys obtained some degree of funding for assessment of micronutrient biomarkers, most of the discussion concerning financial barriers revolved around barriers associated with limitations in available funding. Informants from all six surveys reported lack of sufficient funds to cover the cost of assessing all desired micronutrient biomarkers as a major restricting factor in the decision-making process.

According to recent costing from some of the commonly used international laboratories obtained during the interviews, the minimum cost per individual for ferritin, at least one inflammation biomarker, serum retinol, RBC folate, and vitamin B12 was approximately 65 USD —before including urinary iodine, other micronutrient biomarkers, malaria tests needed in malaria endemic environments, costs associated with specimen collection, processing, and sample storage, or shipping costs. Within the available funds, countries did their best to assess as many biomarkers as possible, and they were advised by the external supporting agency on how to do so. The in-country experiences were echoed in the interviews with the external technical support agencies:

“*It’s always, always money*.”

On several occasions, interviews with external support agencies touched on the difficulty in obtaining funding for inclusion of micronutrient biomarkers at all. It was noted that nutrition, let alone monitoring of micronutrient status, was suffering from inadequate financing. Furthermore, field costs in terms of phlebotomists’ salaries, vehicle and field laboratory processing, and cold-chain expenses were additional barriers. It was also noted that, after cost, there was a limited pool of people with the technical expertise (survey design, laboratory analysis, data analysis and interpretation) required to support these surveys globally, which the informant pointed out was also related to the issue of funding. When asked about the biggest obstacle for measuring micronutrient biomarkers in a national survey again, one informant said without hesitation:

“*Funding. It is always a problem. People aren’t really interested in [funding] surveys anymore*.”

#### 3.3.2. Lack of Knowledge/Awareness

A recurrent theme across interviews with the external support agencies was a lack of awareness among in-country decision makers and within donor and development partner agencies about the usefulness of micronutrient biomarker data for justifying, designing, and targeting programs. One external support agency informant emphasized that, during some survey design deliberations, the topic of micronutrient biomarkers was not mentioned at all as a priority by country stakeholders involved in the DHS. In one country, the survey investigator came to realize that there was limited knowledge about micronutrient biomarkers among stakeholders, even at the local universities, which were more focused on other aspects of undernutrition such as child wasting or stunting. A representative for the lead external support agency to the government for another country mentioned country-level accountability as a barrier, rather than a motivating factor, to fundraise for micronutrient biomarkers.

#### 3.3.3. Laboratory Analysis-Related Barriers

Overall, country representatives reported a desire to carry out all laboratory analyses domestically, while acknowledging external advice regarding the large cost involved in developing and maintaining high-quality laboratory capacity. Ultimately, the lack of a field-friendly, microvolume, multiplex analytical method was reported by all interviewees as a major barrier for frequent, large-scale micronutrient biomarker assessment. Such methods would enable in-country analysis without considerable investments and circumvent any national policies prohibiting exportation of samples.

In lieu of point-of-collection methods and with limited in-country laboratory capacity, some countries shipped samples to up to four different labs globally. This complicated logistics (delays associated with import permits and/or contractual agreements) and added cost to the survey, and survey investigators had to defend this strategy to stakeholders within the country. Once deciding to ship the samples, the barrier appeared to be a limited pool of laboratories that could analyze multiple biomarkers, including less commonly assessed micronutrient biomarkers, at acceptable quality and reasonable cost.

#### 3.3.4. Contextual Barriers

Every survey had its own unique circumstances and contexts. In Uzbekistan, the prohibition of exporting human biological samples, combined with limited capacity of domestic laboratories, meant that it was only possible to include micronutrient biomarkers for which there was domestic capacity for analysis. Ghana was unique in having a very short timeline for completing the survey. This time pressure, combined with funding restrictions, prevented the inclusion of micronutrient biomarkers that were more complicated to collect, process, or measure, such as plasma/serum zinc. In Uganda, the panel of micronutrient biomarkers ultimately included in the survey was limited because they were piloting the integration of micronutrient biomarker assessment in the Uganda National Panel Survey (UNPS).

In two of the surveys, a broader panel of micronutrient biomarkers were perceived to be in “competition” with the desired level of representativeness for the survey. In other words, despite smaller sample sizes being required for biochemical assessment of micronutrient status compared with other indicators of nutritional status (e.g., anthropometry), larger sample sizes were required when representativeness beyond urban/rural or macro-regions was desired. For example, the Uzbekistan survey required results that were representative at the level of 13 oblasts (regions), and the survey in Pakistan was carried out with district-level representativeness:

“*When you survey 115,000 households from every district in Pakistan you have to be parsimonious in terms of what you can feasibly do and justify*.”

### 3.4. Enablers of Inclusion of Micronutrient Biomarkers

#### 3.4.1. Government Support and Commitment

All key informants stressed the importance of support and commitment from government officials, whose motivation frequently originated from a desire to assess the success of their programs or to address a certain health condition where a micronutrient was implicated. Herein laid an implicit enabler once government support and commitment were present: micronutrient biomarkers whose micronutrients were associated with specific public health programs (e.g., vitamin A supplementation, salt iodization, etc.) were always prioritized for inclusion in the survey.

In Malawi, two previous micronutrient surveys “informed us greatly in terms of how we are doing”, and this previous experience was clearly a motivating factor to measure whether salt fortification with iodine or interventions with iron and vitamin A were “making a difference in the people’s lives”. The government also wanted to understand what was driving the continued high prevalence of stunting and the high birth defect rates despite a policy of iron–folic acid supplementation during pregnancy. This commitment enabled the measurement of plasma/serum zinc, folate, and vitamin B12 in addition to biomarkers of iron status.

In Cambodia, the driving force for the government was to investigate the causes of the high prevalence of anemia, which could not be resolved by provision of iron supplements. The government gave autonomy to IRD and UNICEF to add more micronutrient biomarkers to the survey as long as this question was answered.

Echoing the country interviews, the most important enabler cited by CDC and UNICEF informants was having in-country support and government commitment. In line with this, the CDC informant stressed the importance of having a local survey organization with expertise in implementing population-based surveys, and that the local organizers were communicative and responsive.

#### 3.4.2. Advocacy

There were several examples of the importance of in-country advocates. In Pakistan, focused advocacy by lead academics toward a specific donor over several years resulted in securing funding for the survey. At the time of the Malawi survey, the Ministry of Health’s Department of Nutrition was in a position of influence, directly under the office of the President’s cabinet. In addition, donors were highly motivated to capitalize on the unique and potentially cost-effective opportunity of including micronutrient assessment in a DHS survey. In Cambodia, IRD and UNICEF had recent experience with assessment of micronutrient biomarkers in a similar context (Vietnam) and could capitalize on the government’s drive to understand the identified anemia problem.

#### 3.4.3. Cost Savings on Laboratory Analyses

Being able to make savings on the laboratory analysis costs was an enabling factor. In Pakistan, all biomarkers were analyzed in-country and at-cost in the laboratory at Aga Khan University. Similarly, the survey in Cambodia could make use of regional laboratories and one domestic laboratory for several analyses, which also reduced the cost.

Savings were also realized by using contract laboratories that could carry out many analyses, thereby preventing multiple shipments. The VitMin Laboratory was used by all four surveys where samples were exported because of its ability to measure five biomarkers at an exceptionally low price and with a minimal volume of serum (100 μL).

Other approaches to reduce costs of laboratory analyses were assessing expensive biomarkers, such as serum retinol and MRDR, on a subsample (see Table 2) or by only analyzing certain biomarkers only for the population group for which it was considered most critical, e.g., RBC/serum folate for WRA only.

#### 3.4.4. Nutrition as a Priority

The UNICEF informant stressed the importance of designating nutrition, inclusive of micronutrients, as a priority, using the first Comprehensive National Nutrition Survey in India as an example. UNICEF advocated that undertaking a comprehensive national survey was a priority through documentation of the paucity of robust data from previous surveys, which could not be used to make national or state-level estimates of micronutrient deficiencies. In addition, “everyone kept talking about it” from a programmatic perspective, emphasizing that if they knew what they were facing, they could better know how to adjust funding demands and program priorities, and that there was a clear cost–benefit in this. Throughout the process, there was strong national ownership among the Indian academic community, taking leading roles in analysis and advocacy for policy based on the results.

### 3.5. Challenges during Survey Implementation and Their Solutions

The country survey teams were by-and-large prepared for the challenges they might face, as quoted: “that’s what you get from experience”. In some cases (e.g., Pakistan), sufficient experience was present in-country, and, in many instances, external technical support agencies (e.g., CDC and GroundWork) provided additional technical assistance.

The challenges faced during data collection, processing, transport, and storage are summarized in Table 4. Ensuring that the cold chain was maintained despite intermittent availability of electricity was mentioned as the first or second challenge for nearly all surveys. However, all countries were well prepared with necessary mobile solutions. In the surveys where micronutrient biomarker data collection followed a DHS survey (Cambodia, Malawi) or was included in the National Panel Survey (Uganda), the coordination with the main survey team was reported as challenging with several lessons learnt for the next time around. Obtaining venous blood samples was reported as a major issue for two surveys, either because of misperceptions or refusals, or because it was added to an existing surveillance system that previously only used fingerstick sampling.

### 3.6. Using Existing Survey Platforms to Collect Micronutrient Biomarkers: Experiences from Cambodia, Malawi, and Uganda

#### 3.6.1. Country Perspectives

Cambodia and Malawi trialed combining a micronutrient survey with a DHS. In addition, Uganda piloted micronutrient biomarker assessment as part of a national surveillance system. The country motivation to “marry a micronutrient survey with a DHS” was primarily to learn whether it could be a cost-effective way of obtaining nationally representative micronutrient status data, as well as for the opportunity to link these data with a larger, more comprehensive dataset. Coordination challenges were experienced in both countries, but Malawi, in particular, indicated that they would conduct a micronutrient survey in association with a DHS again. However, they would “need to start the discussions at the same time and have stakeholder agreement from the start”. Informants for Cambodia and Malawi said they would strongly prefer integrated data collection, including visiting the clusters at the same time and accessing DHS electronic forms.

Uganda piloted using a national surveillance system, the UNPS, to collect nationally representative data on micronutrient status. The key informants considered it a relatively cost-effective model; compared to a standalone survey, the micronutrient biomarker component could utilize some of the same staff and logistical structures. In addition to the flexibility of the platform, which allowed phasing of the collection of micronutrient biomarkers, the unique advantage of the UNPS was the assessment of longitudinal trends, with each household being followed for 10 years. A major disadvantage was the relatively high burden placed on households year after year. As for Cambodia and Malawi, coordination between survey teams was a challenge.

#### 3.6.2. Technical Support Agencies’ Perspectives

The ICF Macro informant reported that countries were increasingly requesting micronutrient biomarker surveys to be combined with DHS surveys because the national statistics offices were overstretched. The concern with this model from the perspective of the DHS Program was that adding micronutrient status assessment would overburden the traditional survey and, in turn, compromise the overall data quality. In addition, having a cold chain is not part of the regular procedures for the core data collected in DHS surveys.

The supporting agencies noted that a major barrier of this model was laboratory analysis. Countries had a strong ownership of their DHS survey, including a desire to carry out laboratory analysis in-country, and ICF Macro’s mandate was to provide technical assistance to countries so they could eventually carry out surveys on their own. However, the ICF Macro informant noted that, if countries were amenable to shipment of samples, ICF Macro would have no objections.

Regarding potential cost-effectiveness, the ICF Macro informant reiterated that the main advantage of a combined survey is the unified platform and dataset, not the (likely small) cost savings, and that, ultimately, the overarching barriers to including micronutrient biomarker assessment in DHS surveys are inadequate funding to perform the micronutrient biomarker analyses of interest and the risk of overburdening the DHS survey.

## 4. Discussion

This study captured the perspectives of key in-country and external agency informants involved in six national-level assessments of micronutrient status. Each country had a unique set of circumstances, and every survey was different. However, some universal barriers to and enablers of the inclusion of biomarkers of micronutrient status were identified, and interviews with informants from key global supporting agencies confirmed and added context to the country experiences.

Reliable, high-quality population-level data on micronutrient status are needed to define whether a deficiency problem exists in a particular population and whether the prevalence is of a magnitude warranting a public health intervention, to identify the subgroups most affected by the deficiency to allow appropriate targeting, to monitor the impact of programs and any adverse effects, and for research purposes to assess the relationship between micronutrient status and health outcomes [3]. Taking Guatemala as an example, vitamin A deficiency was identified as a serious public health problem affecting several population groups and most severely PSC, which prompted the design and implementation of a national sugar fortification program [19]. In recent years, data on improved vitamin A status in Guatemala allowed the scaling back of vitamin A supplementation, reducing the risk of excessive intake and lowering program costs [20]. Similarly, vitamin A biomarker data from the 2015–2016 Malawi Micronutrient Survey included in this study showed the virtual elimination of vitamin A deficiency in Malawi and also raised concerns about vitamin A excess, with serum retinyl esters being elevated in nearly one in five preschool and school-aged children. These results highlighted the need to modify present vitamin A interventions [21].

The lack of funding for laboratory analyses was reported as the most important barrier to the inclusion of micronutrient biomarkers in national surveys and surveillance systems. This was in part ascribed to a lack of awareness about the importance of the data to inform decision making. For donors and policymakers to appreciate the need for the data, more cohesive advocacy, including supporting in-country champions and sharing success stories, is needed. This need for systematic advocacy and awareness raising, as well as the establishment of a multi-donor, central fund to support data collection and laboratory analysis, was recently further articulated by a working group of experts convened by the Micronutrient Forum [3]. The working group also recommended the establishment of regional resource laboratories to receive and analyze biological specimens collected in national surveys, and to provide laboratory training. This would go a long way to streamlining the process, allowing greater scale, and potentially making laboratory analyses more affordable for countries. In the long-term, however, country representatives and external agencies alike agreed that field-friendly multi-analyte instruments/methods would be a major game changer, and that efforts in this area should be intensified. Indeed, being able to assess hemoglobin using the HemoCue device in the field or utilizing a laboratory that could analyze five biomarkers in a small amount of serum at an exceptionally low price enabled assessment of some biomarkers in our study.

Only around four national micronutrient surveys are conducted every year compared with approximately 21 nationally representative health surveys supported by DHS or UNICEF [3], reinforcing the potential advantage of utilizing existing platforms for national-level micronutrient status assessments. The reported experience of Cambodia and Malawi using the DHS platform corresponded well with a more comprehensive evaluation of the Malawi experience [22]; more integration is needed, and planning for integration must start well in advance of the field work. However, the concerns expressed by DHS about the potential negative impact on the quality of the overall survey data require further attention, as do the issues of country-based laboratory analysis and the costs of building and maintaining laboratory capacity in each country.

## 5. Conclusions

A vast data gap exists for characterizing micronutrient status and the prevalence of micronutrient deficiencies in LMICs, and this dearth of information is blocking recognition of the need for and design and targeting of appropriate interventions, as well as tracking of program outcomes and accountability. However, assessment of biomarkers of micronutrient status is expensive, and the major barrier for countries is insufficient funds to pay for the analyses. Advocacy to establish a funding mechanism specifically for micronutrient biomarker assessment, including the development of global contract laboratories, is needed as a first step to increase the availability of high-quality data on micronutrient status in LMICs and, ultimately, improve population health.

## Figures and Tables

**Table 1 nutrients-14-02009-t001:** Characteristics of the six surveys assessing biomarkers of micronutrient status for which key informant interviews were conducted.

	Cambodia 2014	Malawi 2015–2016	Pakistan 2019	Uganda 2018	Ghana 2017	Uzbekistan 2017
Domestic lead agency	UNICEF	Ministry of Health ^2^, NSO ^2^	Ministry of Health, Aga Khan University ^1^	Ministry of Health ^2^, UBOS	University of Ghana ^2^	UNICEF ^2^
External agency	IRD ^1^	CDC ^3^, Emory University	UNICEF, WFP	CDC ^3^	GroundWork ^3^	GroundWork ^3^
Funding sources	UNICEF, World Vision, IRD, WFP, ILSI	Irish Aid, World Bank, UNICEF, USAID	FCDO, USAID, Australian Aid	USAID, UNICEF,	UNICEF, Global Affairs Canada	UNICEF
Platform	“Follow-on” from DHS	“Follow-on” from DHS	Broader nutrition survey	National Panel Survey	Broader nutrition survey	Broader nutrition survey
Representativeness	National Urban–rural	National Urban–rural Regional: 3 zones	National Province District	National Urban–rural Regional: 5 regions	National Urban–rural Regional: 3 zones	National Regional: 14 regions
Micronutrient biomarkers assessed	Hemoglobin Ferritin sTfR RBP U. iodine S. folate Vitamin B12 Zinc Vitamin D Calcium RBC ThDP CRP AGP	Hemoglobin Ferritin sTfR Retinol RBP MRDR U. iodine S. folate RBC Folate Vitamin B12 Zinc Selenium CRP AGP	Hemoglobin Ferritin Retinol U. iodine S. folate RBC folate Vitamin B12 Zinc Vitamin D Calcium CRP AGP	Hemoglobin Ferritin Retinol RBP MRDR U. iodine S. folate RBC Folate Vitamin B12 CRP AGP	Hemoglobin Ferritin sTfR Retinol RBP MRDR S. folate Vitamin B12 CRP AGP	Hemoglobin Ferritin Retinol U. iodine S. folate Vitamin B12 CRP AGP
*N* PSC	801	1500	31,828	1600	1234	2277
*N* WRA	725	780	31,828	2300	1216	2520
Laboratories used for analysis	2 international 2 regional 1 domestic	4 international 1 domestic	1 domestic	2 international	3 international	3 domestic

^1^ Key informant represented both domestic lead agency and external technical support agency; ^2^ domestic lead agency key informant; ^3^ external technical support agency key informant. Agencies: IRD, Institut de Recherche pour le Développement; WFP, World Food Program; ILSI, International Life Sciences Institute; DHS, Demographic and Health Survey; NSO, National Statistical Office; CDC, Centers for Disease Control and Prevention; FCDO, Foreign, Commonwealth and Development Office (FCDO) United Kingdom (formerly DFID, Department for International Development); UBOS, Uganda National Bureau of Statistics. Micronutrient biomarkers: U., urinary; S., serum; sTfR, soluble transferrin receptor; RBP, retinol-binding protein; MRDR, modified relative dose response; vitamin D, 25-hydroxy-vitamin D; RBC, red blood cell; ThDP, thiamine diphosphate; CRP, C-reactive protein; AGP, α-1-acid glycoprotein. Other abbreviations: PSC, preschool children; WRA, women of reproductive age.

**Table 2 nutrients-14-02009-t002:** Micronutrient biomarkers assessed in selected population groups in the six surveys for which key informant interviews were conducted.

		Cambodia		Malawi		Pakistan		Uganda		Ghana		Uzbekistan	
		PSC	WRA	PSC ^1^	WRA	PSC	WRA	PSC	WRA	PSC	WRA	PSC	WRA
**Iron**	Hemoglobin	☑	☑	☑	☑	☑ ^3^	☑	☑	☑	☑	☑ ^5^	☑	☑ ^5^
	S. ferritin	☑	☑	☑	☑	☑	☑	☑	☑	☑	☑	☑	☑
	S. sTfR	☑	☑	☑	☑			☑	☑	☑	☑		
**Vitamin A**	S. retinol			☑ ^2^	☑ ^2^	☑	☑	☑ ^2^	☑ ^2^	☑ ^2^	☑ ^2^	☑	☑
	S. RBP	☑	☑	☑	☑			☑	☑	☑	☑		
	MRDR			☑ ^2^	☑ ^2^			☑ ^2^	☑ ^2^	☑ ^2^	☑ ^2^		
**Iodine**	U. iodine	☑	☑		☑	☑ ^4^	☑		☑				☑ ^5^
**Zinc**	P./S. zinc	☑	☑	☑	☑	☑	☑						
**Vitamin D**	P./S. vitamin D		☑			☑	☑						
**Calcium**	S. calcium		☑			☑	☑						
**Folate**	S. folate	☑	☑	☑	☑	☑	☑		☑		☑		☑
	RBC folate				☑	☑	☑		☑				
**Vitamin B12**	P./S. vitamin B12	☑	☑	☑	☑	☑	☑	☑	☑		☑		☑
**Thiamine**	RBC ThDP	☑	☑										
**Selenium**	P. selenium			☑	☑								
**Inflammation**	S. CRP	☑	☑	☑	☑	☑	☑	☑	☑	☑	☑	☑	☑
	S. AGP	☑	☑	☑	☑	☑	☑	☑	☑	☑	☑	☑	☑

☑, biomarker was included the survey; PSC, preschool children; WRA, women of reproductive age; S., serum; P., plasma; U., urinary; sTfR, soluble transferrin receptor; RBP, retinol-binding protein; MRDR, modified relative dose response; vitamin D, 25-hydroxy-vitamin D; RBC, red blood cell; ThDP, thiamine diphosphate; CRP, C-reactive protein; AGP, α-1-acid glycoprotein. ^1^ The same micronutrient biomarkers were also assessed in school-aged children; ^2^ measured in a subsample; ^3^ also assessed in adolescent girls; ^4^ urinary iodine only for children 6–12 years; ^5^ also assessed in pregnant women.

**Table 3 nutrients-14-02009-t003:** Importance and frequency of reported barriers to inclusion of micronutrient biomarkers in national surveys and surveillance systems in six countries.

	High Frequency	Medium Frequency	Low Frequency
High importance	Financial: Difficult to obtain funding. Available funding not adequate to cover the analysis cost for all micronutrient biomarkers. High laboratory analysis costs. Limited human resources globally to support countries. ^1^ Programmatic: Micronutrient not associated with a program.	Laboratory: Lack of laboratories capable of multiple analyses, within available resources. Lack of field-friendly multiplex methods. ^1^ Awareness/knowledge: Lack of awareness of the need for the data among donors, development partners, or governments. ^1^	Contextual: Export of blood samples prohibited. Limited time available to conduct survey. Micronutrient biomarkers ‘competing’ against the level of representativeness of the survey. Laboratory: Domestic laboratory capacity dictated which biomarkers were selected. Lack of available laboratories for some micronutrients.
Medium importance	Awareness/knowledge: Government concerned survey would put them in a bad light or that they would be held accountable. Lack of understanding of the need to analyze samples abroad or of analysis quality.		Other technical concerns: Complexities in collection for some micronutrients.

^1^ Barriers emphasized by representatives from external supporting agencies.

**Table 4 nutrients-14-02009-t004:** Main challenges faced during collection, processing, transport, and storage of blood and urine samples.

Challenge	Frequency	Solutions and Enabling Factors
Ensuring cold chain was maintained	5	Cold boxes, freezer in car, car battery for mobile freezers, mobile field labs or district hospitals for processing and/or temporary storage Training and supervision of field staff by lab technicians
Coordination with DHS or panel survey	3	Active coordination and frequent communication Leadership and support from government authorities Gathering all teams and agreeing on a plan
Obtaining the blood sample (major issue in 2 surveys)	2	Establishing rapport with study population Competent and experienced staff
Long lead times for supplies (major issue in 2 surveys)	2	Allowing long timelines, circumventing UNICEF Supply Division for specific supplies
Obtaining approvals for shipment of samples	2	Lesson learnt: factoring in time and resources for this
Rainy season, snow, security challenges	2	Determination and perseverance
Short duration of training and varying quality of staff hired	2	Hiring field workers with nutrition/health background

## Data Availability

Not applicable.

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
