# Peer review of "Barriers to and Enablers of the Inclusion of Micronutrient Biomarkers in National Surveys and Surveillance Systems in Low- and Middle-Income Countries"

_nutrients, 2022, doi:10.3390/nu14102009_

Round 1
Reviewer 1 Report
- Page 2: line 49-51: “With the caveat that more surveys may have been conducted, but results were not made available, between 1988 and 2018, just 38% of LMICs reported data for PSC”. please provide specific LMICs (i.e., individual LMI countries) that reported data. I am not sure what ‘PSC’ refers to!
- Page 2: line 53-54: “Data on thiamine, riboflavin or selenium status among PSC have only been generated by a handful of LMICs”. Clarify “handful of LMICs” with references.
- Too many acronyms being used in the manuscript without describing them in the first place. For example, IZiNCG, PSC, ICF, DNHA, etc. Please include a list of acronyms.
- Table 2 overlaps with Table 1, and Table 2 includes country surveys, surveillance systems and key informant affiliations under the sub-category "Domestic lead agency" and "external agency". Therefore, please remove Table 1.
- Table 2: please provide the number of participants included in these six surveys (survey-specific numbers)
- Page 8, Lines 254-255: “Funding. It is always a problem. People aren’t really interested in [funding] surveys anymore.” why? and what are the reasons?
- Table 5: what do you mean by "keep going" in the presence of challenges such as snow, security and rainy season?
Author Response
Responses to reviewer 1 nutrients-1660930
- Page 2: line 49-51: “With the caveat that more surveys may have been conducted, but results were not made available, between 1988 and 2018, just 38% of LMICs reported data for PSC”. please provide specific LMICs (i.e., individual LMI countries) that reported data. I am not sure what ‘PSC’ refers to!
Response: Thank you for picking up the error that PSC had not been defined. The abbreviation stands for preschool children, and this description has now been included in line 51. We appreciate that listing the individual LMICs that have data on micronutrient biomarkers would be useful. However, this is not feasible given the large number of LMICs. As an example, “38% of LMICs” is 53 countries. We refer you to the WHO Micronutrient Database located on the Vitamin and Mineral Nutrition Information System website for a comprehensive list of published nationally representative micronutrient biomarker data (https://www.who.int/teams/nutrition-and-food-safety/databases/vitamin-and-mineral-nutrition-information-system).
- Page 2: line 53-54: “Data on thiamine, riboflavin or selenium status among PSC have only been generated by a handful of LMICs”. Clarify “handful of LMICs” with references.
Response: This sentence has been modified to “Very few LMICs have published data on thiamine, riboflavin or selenium status among PSC and WRA” to indicate that we are referring to published results (lines 54-55). References have also been added. It is a challenge that not all survey findings are published, or are significantly delayed in being published.
- Too many acronyms being used in the manuscript without describing them in the first place. For example, IZiNCG, PSC, ICF, DNHA, etc. Please include a list of acronyms.
Response: Thank you for noticing that PSC and IZiNCG had not been described when first mentioned. This has been corrected on lines 51 and 82, respectively. ICF is a historical acronym for “Inner City Fund”, and it is ICF Macro’s custom and preference not to define the acronym. DNHA was only mentioned once, and we have therefore removed the acronym (line 363). We have not included a list of acronyms but ensured that all acronyms have been defined the first time they are used, in addition to the list of acronyms that are indicated in the footnotes of the tables.
- Table 2 overlaps with Table 1, and Table 2 includes country surveys, surveillance systems and key informant affiliations under the sub-category "Domestic lead agency" and "external agency". Therefore, please remove Table 1.
Response: Thank you for pointing out the overlap. Table 1 has been removed and Table numbers have been updated accordingly. With removal of the “old” Table 1, the “new” Table 1 contains footnotes to indicate from which agencies the key informants belonged. We have also added some clarifications in the beginning of the results section (lines 142-149).
- Table 2: please provide the number of participants included in these six surveys (survey-specific numbers)
Response: Thank you for this suggestion. The number of participants included in these surveys for which micronutrient status was assessed have been added in a row below “Micronutrient biomarkers assessed” in the new version of Table 1. Please note that for consistency, the n for preschool children and women of reproductive age combined is reported, as these population groups were included in all surveys.
- Page 8, Lines 254-255: “Funding. It is always a problem. People aren’t really interested in [funding] surveys anymore.” why? and what are the reasons?
Response: This is a direct quote from an interviewee who did not provide further reasons. Respondents from the external technical support agencies talked in broader terms about the area of nutrition being underfunded, and micronutrient programs and data collection on micronutrient status being in competition with other public health programs.
- Table 5: what do you mean by "keep going" in the presence of challenges such as snow, security and rainy season?
Response: “keep going” has been replaced by “determination and perseverance”. We hope this is more descriptive of the solution / enabler in this situation.
Reviewer 2 Report
The current manuscript drives the hypothesis of the developemnt and funding acquisition of a biomarker nutrient and micro-nutrient data bank available for the agencies and consumers especially for the low and middle- income countries, so the population health can be improved. The article is well written and prepared and is based on the data derived from 6 countries. In my opinion this survey has a scientific value and could potentially aid to the purpose of its presentation. The English language is very good and minor issues are needed revision such as the citing format of the references used. A major issue is the lack of statistical analysis section and statistical analysis data interpretation. A discriminant analysis for this countries related to the measured/dicussed independent parameters would add further scientific value. I have indicated this problem within the attached pdf. My opinion about this survey is in general positive (minor to major revision).

Author Response
Responses to Reviewer 2 nutrients-1660930
Comment from Reviewer 2 in online form:
The current manuscript drives the hypothesis of the development and funding acquisition of a biomarker nutrient and micro-nutrient data bank available for the agencies and consumers especially for the low and middle- income countries, so the population health can be improved. The article is well written and prepared and is based on the data derived from 6 countries. In my opinion this survey has a scientific value and could potentially aid to the purpose of its presentation. The English language is very good and minor issues are needed revision such as the citing format of the references used. A major issue is the lack of statistical analysis section and statistical analysis data interpretation. A discriminant analysis for these countries related to the measured/discussed independent parameters would add further scientific value. I have indicated this problem within the attached pdf. My opinion about this survey is in general positive (minor to major revision).
Response: Thank you for your comment about the value of this study. Regarding your comment about statistical analysis, such as discriminant analysis, we would like to reiterate that this study was designed as a qualitative study following a thematic analysis approach and was not intended for statistical analysis.
Comments from Reviewer 2 in PDF of manuscript:
- The cited references should be in brackets.
Response: Thank you for pointing out this omission; the cited references are now in brackets.
- Materials and methods: Statistical analysis section is required; Results: Statistical analysis data is missing.
Response: As stated above, this study was designed as a qualitative study. We only collected qualitative data from in-depth interviews and did not collect quantitative data that would lend itself to traditional statistical analyses. As described in our Methods section, we used a thematic analysis approach as recommended for qualitative interview data.
Round 2
Reviewer 2 Report
The revised version of the manuscript has been improved. The authors justified why they did not include statistical analysis. In this context, I suggest the publication of their study in its current form.